# DELVING INTO LLMS' VISUAL UNDERSTANDING ABILITY USING SVG TO BRIDGE IMAGE AND TEXT

## ABSTRACT

Large language models (LLMs) have made significant advancements in natural language understanding. However, through that enormous semantic representation that the LLM has learnt, is it somehow possible for it to understand images as well? This work investigates this question. To enable the LLM to process images, we convert them into a representation given by Scalable Vector Graphics (SVG). To study what the LLM can do with this XML-based textual description of images, we test the LLM on three broad computer vision tasks: visual reasoning, image classification under distribution shift, and generating new images using visual prompting. Even though we do not naturally associate LLMs with any visual understanding capabilities, our results indicate that the LLM can indeed do a pretty decent job in many of these tasks, potentially opening new avenues for research into LLMs ability to understand images.

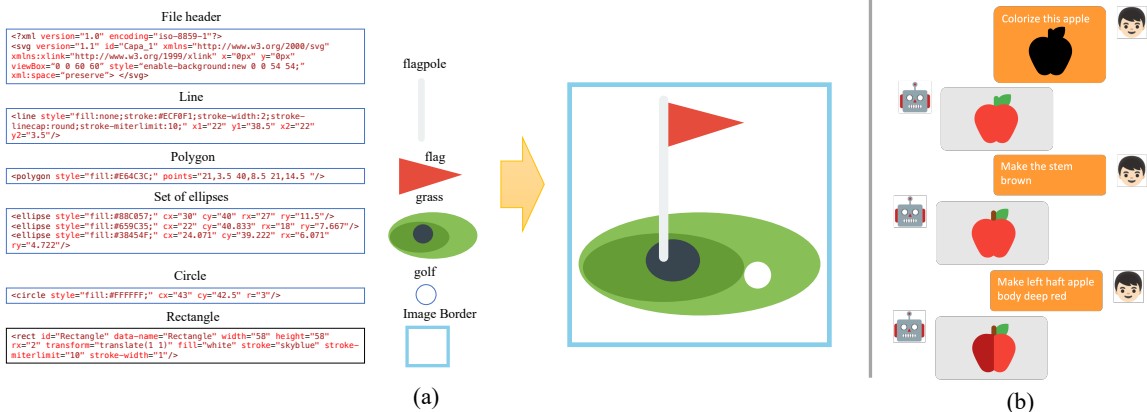

Figure 1: (a) An SVG representation illustrating a golf course. Each geometric shape represents a distinct object. (b) LLMs are able to understand and generate shapes, color, and relationships between different elements in an interactive manner.

## 1 INTRODUCTION

Large-scale data and enormous compute: the effect of these two ingredients has been on display in recent years in the significantly increased capability of machine learning systems. Models operating on the two most popular forms of data - image and text - have particularly felt that effect the most. From the side dealing with textual data, we have seen the emergence of large language models (LLMs) such as Chat-

GPT (OpenAI, 2023a) and GPT-4 (OpenAI, 2023b). Similarly, the vision side has also shown impressive accomplishments (Dosovitskiy et al., 2020; Liu et al., 2022; Dehghani et al., 2023).

However, when we compare these two realms, the abilities of LLMs do stand out in a distinct way because of their remarkable abilities in reasoning, in-context learning, and open-ended tasks (Bubeck et al., 2023). These analytical capabilities are something that the vision models, despite their significant advances, have not yet mirrored to the same depth (Dehghani et al., 2023; Alayrac et al., 2022).

This distinction can be attributed to the inherent nature of their respective data: LLMs thrive on the diverse and sequential structure of textual data, which is conducive to understanding intricate relationships and producing contextually relevant responses. In contrast, the continuous and varied nature of visual data complicates the discernment of nuanced relationships, potentially hindering the depth of analysis LVMs can achieve (Bar et al., 2022; Diwan et al., 2022; Lake et al., 2017). Moreover, there is an ongoing debate on whether LLMs, trained on internet-scale data, can learn world models that could lead to AGI capabilities, or that they are fundamentally limited due to their lack of grounding on physical interaction and visual perception in the real world (LeCun, 2022).

The disparity between LLMs and LVMs, and the debate on the necessity of physical interaction and perceptual grounding, intrigue us to a question: *Can LLMs, which have never seen visual data, understand and reason about images?* Answering this question will bring us closer to understanding the capabilities of LLMs beyond the textual domain, whether they possess world models, and what their fundamental limitations may be. As such, our study takes a small but important step toward this goal.

In order to enable an off-the-shelf pre-trained LLM to "read" images, we use the Scalable Vector Graphics (SVG) (Ferraiolo et al., 2000) representation to convert images into readable text. Unlike traditional pixel-based images, SVGs are described in XML, offering a text-based portrayal of mid-level shapes, curves, lines, and colors, as shown in Figure 1. The textual nature of SVG provides a data modality that LLMs excel at, acting as a bridge to apply their analytical strengths to the visual domain. While the Sparks of AGI paper (Bubeck et al., 2023) showed some initial qualitative results on the image understanding capabilities of LLMs using a similar idea, we provide a deeper, comprehensive study that includes both qualitative and quantitative analyses on a variety of visual understanding and reasoning tasks.

Specifically, we evaluate whether an LLM can perform both discriminative and generative visual understanding tasks. For discriminative tasks, we study their visual reasoning capability as well as few-shot in-context learning performance for image classification tasks, and measure their robustness to distribution shifts in visual data. Surprisingly, despite never having seen dense visual data, LLMs perform much better than chance and are often robust to distribution shifts. For generative tasks, we study LLMs' image generation and editing capabilities based on interactive, chat-based feedback. We find that LLMs can identify and execute transformations related to color, shape, style, and content within the SVG image representation, to generate credible outcomes.

## 2 RELATED WORK

### 2.1 LEVERAGE LLMs FOR VISUAL TASKS

Upon observing the powerful reasoning capabilities of LLM, researchers began to harness its potential for visual tasks. Presently, there are three primary approaches to utilizing LLM for these purposes: 1. The first approach involves using LLM to produce textual guidelines. Vision models then rely on these instructions to execute a range of visual tasks. Examples include Visual ChatGPT (Wu et al., 2023), visual programming as seen in Gupta & Kembhavi (2023), and ViperGPT (Surís et al., 2023). 2. The second approach, as illustrated by LLaVa (Liu et al., 2023) and MiniGPT4 (Zhu et al., 2023), incorporates the pretrained vision encoder model, along with a trainable linear projector. This allows for feeding visual features directly to LLMs, demonstrating remarkable reasoning abilities. VisionLLM (Wang et al., 2023) presents bounding boxes

and segmentation masks in text format (as polygons), enabling LLMs to address more intricate perception challenges. 3. The third approach seeks to represent images directly in a text-based format, bypassing the use of visual encoders. The goal here is to allow LLMs to interpret these text-based representations. For instance, LIFT (Dinh et al., 2022) represents images using their raw pixel values in textual form and then fine-tunes the language model on them for visual tasks. Another study (Bubeck et al., 2023) explores image generation by expressing the image in text formats, like TiKZ or SVG.

**Key difference:** Our research aligns with the third approach. Unlike LIFT (Dinh et al., 2022), we represent images using SVG, a format that inherently encodes more structural information than raw pixel values. This could enable LLMs to better grasp intricate relationships and yield contextually relevant responses. Distinct from the methods presented in Bubeck et al. (2023), we conduct a comprehensive study of how LLMs process images via textual representations, including both discriminative and generative tasks.

## 2.2 SCALABLE VECTOR GRAPHICS

Vector graphics describe images as collections of parameterized shape primitives such as polygons, circles, and rectangles, rather than a regular raster grid of pixel values (Peng & Zhang, 2004). This representation is extensively supported by web browsers and can be rendered without any special software or plugins (Badros et al., 2001). Primitives are usually characterized by a set of coordinates delineating their contour and the associated color. This leads to a compact and infinitely scalable representation where the appearance can be easily modified by adjusting stroke or color parameters. Consequently, vector graphics are the preferred choice among graphic artists and designers, as images maintain their sharpness regardless of the zoom level. Encapsulated PostScript (EPS) and Scalable Vector Graphics (SVG) are two notable vector-based formats (Ferraiolo et al., 2000).

SVG format stores images as XML-based text files that define geometrical objects and their properties (Ferraiolo et al., 2000), shown in Figure 1. This enables easy editing, manipulation, and embedding, which makes SVG particularly versatile for web applications and graphic design tasks (Badros et al., 2001). EPS is another vector format for high-quality graphics that can be resized without losing quality (Gruber et al., 2008). In this paper, we employ large language models (LLMs) to understand images in the SVG format, achieving robust shape-color debiasing along with enhanced visual understanding and generation.

## 2.3 LARGE LANGUAGE MODELS

Large Language Models (LLMs) have attracted much attention in recent years due to their remarkable performance across numerous natural language processing tasks. GPT-3 (Brown et al., 2020a), developed by OpenAI, is a prime example of this category, boasting an immense scale of 175 billion parameters and human-like text generation capabilities. In a similar vein, BERT (Devlin et al., 2019) (Bidirectional Encoder Representations from Transformers), introduced by Google, takes advantage of the transformer architecture and has substantially enhanced the state-of-the-art across various tasks by learning deep bidirectional representations. ChatGPT (OpenAI, 2023a), another noteworthy model, is a GPT variant specifically designed for human-like conversational abilities. The most recent iteration, GPT-4 (OpenAI, 2023b), succeeds GPT-3 (Brown et al., 2020b) and carries on the LLM advancements in terms of scale and performance. These models lay the groundwork for our research, enabling us to investigate their potential in more complex tasks such as image processing and understanding. Our work effectively illustrates the applicability of LLMs to SVG-based image understanding and generation, paving the way for novel applications and research directions in the visual domain.

## 3 TASKS AND EXPERIMENTAL RESULTS

The domain of computer vision contains a variety of problems, and often times, models need to have different kinds of abilities to solve them. In this section, we wish to investigate if LLMs can indeed have those required

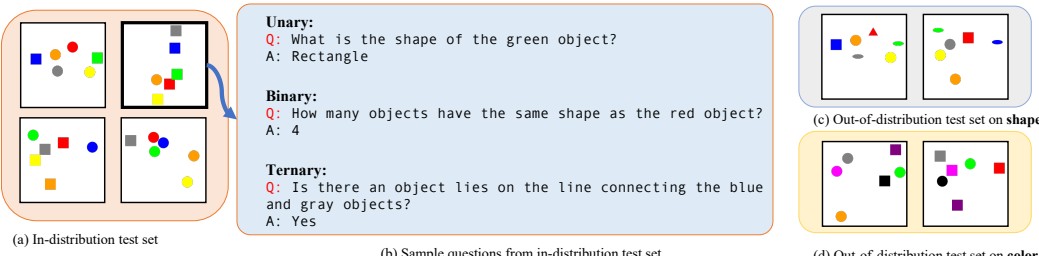

Figure 2: Illustration of In-Distribution and Out-of-Distribution Test Sets: (a) Images from the in-distribution test set, showcasing random sampling of object color, shape, and location. (b) Accompanying each image are questions assessing unary, binary, and ternary reasoning capabilities. (c) Expansion of shape variety to include ellipses alongside rectangles and circles. (d) Introduction of additional colors, including magenta, black, and purple, to the sampling palette.

abilities. However, for them to solve a computer vision problem, there needs to be a way for them to *see* an image. SVG can be that bridge, where an image is converted into a structured XML code (please see Fig. 1 for an example). And like any other code, LLMs can potentially *read* the code and perform some task. To test what is possible using this form of image representation, we consider three broad categories of computer vision tasks.

Sec. 3.1 first studies the problem of visual reasoning, where the model is asked certain kinds of questions about the contents of an input image (e.g., *How many objects have the same shape as the red object?*). Next, in Sec 3.2, we study how LLMs fare in image classification tasks; especially under non-trivial settings like distribution shifts and few-shot learning. After studying their abilities in discriminative tasks, we then test the generative modeling capabilities of LLMs through the task of visual prompting, which asks the model to generate a new image following the pattern depicted by three images - A : B, C : ?.

Unless otherwise mentioned, all the experiments will use GPT-4 OpenAI (2023b) as the LLM model. The common theme across all these different experiments will be that the LLM will be processing all the images converted in the form of SVG.

## 3.1 VISUAL REASONING

With all the successes in the traditional perception tasks like image classification/segmentation Liu et al. (2022); Kirillov et al. (2023), visual reasoning still remains a pivotal challenge for many modern computer vision systems Zhang et al. (2019); Barrett et al. (2018); Santoro et al. (2017). It typically refers to the ability of a model to answer questions about different constituents of an image. To better understand what entails that task, we will discuss the dataset we will be using for this experiment.

**Dataset:** We use Sort-of-Clevr dataset that was introduced by Santoro et al. Santoro et al. (2017). As shown in Figure 2, each image in the dataset is composed of 6 objects, each with a unique color, having a randomly chosen shape between rectangle and circle. For each such image, the dataset contains many {question, answer} pairs. There are three categories of questions crafted to test three levels of reasoning capabilities: unary, binary, and ternary relationships. An example question testing binary reasoning ability is - *What is the shape of the object closest to the gray object?* Please refer to Fig. 2 for more examples of the questions. Each question is, ultimately, a classification problem. The methods are evaluated based on their top-1 accuracy on questions from the test set of the dataset. Please refer to appendix for the details about the test sets used for evaluation.

**Methods:** Broadly speaking, we show the results on two kinds of methods - (i) LLMs which are *not* trained for this task, and are only prompted during inference, and (ii) methods which are trained for this task on the training set of Sort-of-Clevr dataset. Among the first category, we analyze (a) GPT4-brief (OpenAI, 2023b), (b) GPT-CoT (Chain of thought) and (c) LLaVa (Liu et al., 2023). Among the second category, we study two methods: (a) CNN+MLP and (b) Relation Networks (Santoro et al., 2017). The reason we choose to evaluate on this category of trained models is to have an idea of what the upper bound could be; i.e., how difficult the task really is. To evaluate any of the LLMs during inference, we transform the original images into geometric primitives into their SVG format. Then we query the LLM using the following prompt: "Give the following SVG image `<svg>...</svg>`, what is the shape of the red object?" The difference between GPT4-brief and GPT4-CoT is the way we ask the final question: in GPT4-brief, as the name suggests, our final question asks the model to provide the answer briefly, whereas in GPT4-CoT, we explicitly ask the model to break down its reasoning before arriving at the answer (please see the appendix for the exact prompt used to elicit this behavior). By querying a LLM, we obtained answers which were then summarized post human evaluation to determine the final accuracy. Due to costs associated with probing GPT4 models, our evaluation was restricted to 120 examples.

Table 1: Visual reasoning performance of Sort-of-Clevr dataset under in-distribution test set and out-of-distribution test set with shape and color distribution shift.

| Distribution shift | i.i.d. | | | | o.o.d. Shape | | | o.o.d. Color | | |
|---|---|---|---|---|---|---|---|---|---|---|
| Question type | GPT4-brief | GPT4-CoT | CNN | Rel. Net. | GPT4-CoT | CNN | Rel. Net. | GPT4-CoT | CNN | Rel. Net. |
| Image format | SVG | SVG | PNG | PNG | SVG | PNG | PNG | SVG | PNG | PNG |
| Unary | 0.50 | 0.90 | 0.65 | 0.89 | 0.95 | 0.58 | 0.82 | 0.95 | 0.56 | 0.83 |
| Binary | 0.90 | 0.36 | 0.57 | 0.66 | 0.95 | 0.36 | 0.44 | 0.95 | 0.57 | 0.66 |
| Ternary | 0.10 | 0.52 | 0.47 | 0.54 | 0.63 | 0.52 | 0.56 | 0.71 | 0.47 | 0.54 |
| Average | 0.50 | 0.89 | 0.65 | 0.75 | 0.84 | 0.49 | 0.61 | 0.87 | 0.53 | 0.67 |

Table 1 shows the top-1 accuracy of different methods (left: zero-shot inference of LLMs, right: methods trained for the task). Note that the results of LLaVa are shown in the appendix. When looking at the two LLM models which process images in the SVG format, GPT4-brief and GPT-CoT, we can first see that their performance is much higher than chance (many questions in the test set have 5-6 correct answers, thereby reducing the chance performance accuracy; please see the appendix). Furthermore, the performance of GPT-CoT even surpasses the performance of a model explicitly trained for this task. If we take a step back and think once more about the nature of SVG representation (Fig. 1), the best case scenario might be when images from the Sort-of-Clevr dataset have the locations of certain shapes embedded in their XML code. But even if such a nicely structured code is often available to the LLM, to properly be able to reason about questions like the ones described in Fig. 2, the LLM needs to precisely perform many mathematical relational operations - all without ever being told how to do it. From that perspective, the results depict that LLMs might be possessing much complex models already.

**Distribution shift:** Furthermore, to study an even more difficult version of the problem, we test the performance of models in distribution shifts. Specifically, we evaluate under both shape and color distribution shifts. As for the color, we replace 3 colors from the original 6 colors with the new colors. For shape, we randomly enlarge the options to further include the ellipse and triangle, as shown in Figure 2 (c) (d). As a result, each object can sample the shape uniformly from the 4 choices. Importantly, we make sure that all visual reasoning questions can be answered using the original one-hot choices for vision models like CNN-MLP and relation networks.

As shown in Table 1, the LLM model (GPT4-CoT) using the SVG format to process images does not suffer much by any of the newly added complications in the test images (e.g., more shapes added under the shape distribution shift), maintaining its ability to perform the reasoning tasks. Here is what this means in simple

Table 2: Image classification results with vision and language model. We utilized the Mini-MNIST dataset, which comprises 100 images, to evaluate GPT4's ability to understand SVG through both zero-shot and one-shot in-context learning. To evaluate the model's robustness against distribution shift, vision model ConvNeXt and language model Vicuna are finetuned on the MNIST training set, and evaluated on the MNIST test set, CMNIST-A, and CMNIST-B respectively. CMNIST denotes the Colored-MNIST dataset. ICL denotes in-context learning.

| Method | ConvNeXt(fine-tuning) | Vicuna(fine-tuning) | GPT4(Zero-Shot) | GPT4(One-Shot ICL) |
|---|---|---|---|---|
| Image Format | PNG | SVG | SVG | SVG |
| MNIST | 99.5% | 99.1% | 20% | 24% |
| CMNIST-(A) | 79.5% | 95.7% | 16% | 19% |
| CMNIST-(B) | 32.6% | 92.9% | 13% | 20% |

words, as understood through an example: if in the original image (before distribution shift), there was a red circle immediately to the left of a blue rectangle, even after introducing other shapes (e.g., triangles), the LLM can still detect the red circle to the left of that rectangle. This is not something which is trivial, because the models which were explicitly trained on the Sort-of-Clevr dataset do suffer a non-trivial loss in performance; both in color and shape distribution shift. Overall, these results indicate that the internal model used by the LLM is surprisingly effective at tasks that we wouldn't have naturally thought of it being good at.

## 3.2 OUT-OF-DISTRIBUTION GENERALIZATION

To DNNs, innocuous transformations can completely change predictions. This has been reported in various cases such as shifting the image by a few pixels (Azulay & Weiss, 2018), adding a bit of random noise (Hendrycks & Dietterich, 2019) or changing the background, color, or texture (He et al., 2021; Arjovsky et al., 2019; Geirhos et al., 2018) while keeping the shape intact. In this section, we aim to investigate if representing images as SVG could mitigate these issues. Specifically, we study if models learn to rely on the color or the background rather than the actual task (recognizing the shapes).

**Datasets**: We have constructed two variants of the Colored-MNIST dataset to assess model robustness against color and background variations. The first version, termed Colored-MNIST-A, assigns a color of either red or green to the foreground, with each color being selected randomly at an equal likelihood of 50%. In the more challenging second version, dubbed Colored-MNIST-B, both the background and foreground are selected from a color palette that includes black, white, red, blue, and green. The background and foreground colors are always distinct, yielding 20 unique color combinations. Visualization of these Colored-MNIST datasets can be viewed in Figure 3. Furthermore, we utilize the curve tracing algorithm to convert MNIST images into the SVG format. More details can be found in the supplementary materials.

**Task and experimental setting**: In the first setup, we fine-tune the ImageNet pre-trained vision model ConvNeXt (Liu et al., 2022) using PNG images and the pre-trained language model Vicuna using SVG-converted images on MNIST. Subsequent testing is carried out on both Colored-MNIST variants (A) and (B). This setup seeks to examine whether the model can prioritize shape over other features for its predictions. In the second setup, our objective is to explore the potential of harnessing the potent in-context capabilities of Large Language Models (LLMs) to enhance image classification using SVG. To this end, we employ GPT-4 (OpenAI, 2023b) to conduct both zero-shot and in-context learning on MNIST variants. More detail on ConvNeXt and Vicuna fine-tuning, prompting for in context learning can be found in the supplementary materials.

**Results and discussion**: In Table 2, fine-tuning Vicunna with SVG representations has shown promising results on the CMNIST-A and CMNIST-B benchmarks, achieving accuracies of 95.7% and 92.9% respectively.



Figure 3: Illustration of the Out-of-Distribution generalization tasks. We train models on the standard Vanilla MNIST dataset and evaluate them on the more challenging OoD datasets: Colored-MNIST-(A) and Colored-MNIST-(B). The goal is to determine whether models inadvertently prioritize color or background over the primary task of shape recognition.

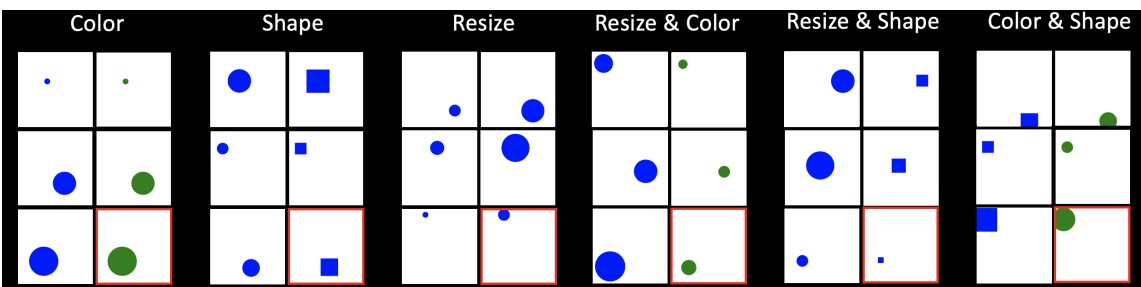

Figure 4: Synthetic data study results. The generation results of our method are annotated with a red square.

This suggests some level of robustness against color and background perturbations. On the other hand, ConvNeXt seems more susceptible to these perturbations, with a noticeable decline in performance on both benchmarks compared to i.i.d results. We hypothesize that SVG might offer a representation more biased towards shape, given its explicit textual encoding of object shapes, allowing for disentanglement of shape from color information. Further, as illustrated in Table 2, there's a notable 4% accuracy boost when using a single in-context sample, as compared to a zero-shot classification approach. This demonstrates the capability of LLM to grasp visual concepts contextually.

### 3.3 VISUAL PROMPTING

The last two sections discussed the emergent abilities of LLMs in discriminative tasks. In this section, we turn our attention towards the generative side, to see if LLMs can understand and generate logically coherent images as well. In particular, we consider the task of visual prompting, where, given a series of images, the goal is to understand the transformation and *fill in* the remaining spot with an appropriate image.

**Dataset**    We follow Bar et al. (2022) to create a set of three simple synthetic tasks of filling in the remaining spot (Fig. 4), and three of their combinations, and evaluate each model on 100 examples per task.

**Tasks and Evaluation.**    Every pair in our example set includes an SVG showcasing a colored shape along with a corresponding SVG with specific transformations. The transformations consist of color, size, or a combination of these aspects. We delve into a more detailed description of each task in the appendix. For evaluation purposes, we adopt the method from Bar et al. (2022), for measuring and reporting the color-aware mean Intersection over Union (mIOU).

Table 3: Synthetic data study results. We report the color-aware mIOU on the six tasks (Bar et al., 2022). It demonstrates that GPT4 is able to understand and reason shape, color, and size transformation using SVG representation.

| Method | Color | Shape | Size | Color Shape | Color Size | Shape Size |
|---|---|---|---|---|---|---|
| VQGAN (Esser et al., 2021) | 7.0 | 19.1 | 16.2 | 7.4 | 2.2 | 18.4 |
| BEiT (Bao et al., 2022) | 40.9 | 31.4 | 7.1 | 33.1 | 21.2 | 13.0 |
| MAE (He et al., 2022) | 70.2 | 44.0 | 34.7 | 19.3 | 19.0 | 46.0 |
| MAE-VQGAN (Bar et al., 2022) | 40.4 | 46.5 | 42.0 | 20.4 | 18.3 | 40.3 |
| SVG with GPT4 | 100.0 | 92.6 | 100.0 | 92.6 | 100.0 | 86.5 |

**Prompt.**    Given two example pairs and a query SVG, we structure the text prompt in the same fashion for all tasks. The prompt is designed to figure out the common transformation in the two examples first and then transform that query into the corresponding key SVG code. We include the prompt details in the appendix.

**Qualitative and quantitative results.**    The results are presented in Table 3. See Figure 4 for our generated results. We believe that GPT4 can clearly understand simple shape, color, and size transformations by analyzing the SVG code without any pixel-level information.

### 3.3.1 STYLE AND CONTENT EXTRAPOLATION

In this section, we assess if LLMs can extrapolate SVG codes with more challenging transformations, such as content and style.

**Style generation:** We present LLMs with sample SVG letters. The first task is to figure out the style in the given examples. Then, given a new test query, the second task is to transform this given query so that it adheres to the same stylistic conventions as the example letters. The qualitative results can be found in the appendix.

**Content generation:** LLMs are shown two examples of SVG code pairs. Each pair consists of a query and key pair (both are numbers), where the query describes an SVG code of a number, and the key describes the SVG code of another number with an introduced mathematical operation. The operation can consist of add, subtract, multiply, and divide. The mathematical operation should be held in both example pairs. The first task is to figure out the mathematical operation in the two examples. Then, given a new test query SVG number, the second task is to identify what number it is and follow the mathematical operation discovered to generate the corresponding test key number. We include qualitative results in Figure 5. The prompt details can be found in the appendix.

## 4 LIMITATION

While our research demonstrates the potential of using Scalable Vector Graphics (SVG) with large language models (LLMs) to tackle visual tasks without a parameterized visual encoder, the major limitation of SVG representation is the loss of fine details: Though our method of converting raster images into SVG format and leveraging XML-based textual descriptions allows for efficient processing of crisp graphics and designs, it is not as effective in handling photographic content. As a result, fine-grained details, such as image textures, may be lost during conversion. Conversely, when the SVG code incorporates an excessive level of detail, its sequence length can become prohibitively long, which can pose challenges for the training and inference of current Transformer-based LLMs. Developing hybrid representations that can retain the advantages of both discrete and continuous data, while preserving finer details, is an area for future exploration. For example, in

| Query | Key | Query | Key | Query | Key | Query | Key | Query | Key | Query | Key |
|-------|-----|-------|-----|-------|-----|-------|-----|-------|-----|-------|-----|
| 12 | 24 | 12 | 24 | 60 | 36 | 60 | 36 | 21 | 14 | 10 | 15 |
| 6 | 12 | 6 | 18 | 25 | 1 | 25 | 15 | 12 | 8 | 20 | 25 |
| 30 | 60 | 30 | 42 | 40 | 16 | 40 | 24 | | 5 | | 30 |

Figure 5: Understanding SVG content through the lens of GPT-4: GPT-4 demonstrates its ability to generate accurate content by analyzing the correlation between provided example number pairs, and subsequently applying this relationship to ascertain the corresponding test key number. Remarkably, in scenarios where the relationship exhibits ambiguity, GPT-4 cna identify multiple possible interpretations.

LLMs, the processing unit is the token, which can correspond to one or several words. However, in SVG, we would prefer to have a specific embedding module for each geometric primitive in SVG, such as circles, polygons, and so on.

Additionally, our empirical tests highlighted certain areas where LLMs fell short, particularly in handling low-level image manipulation tasks. For instance, when prompted to manipulate SVG images in tasks like enlarging dimensions, shrinking dimensions, or rotations, LLMs like GPT-4 displayed inadequate proficiency. Such operations, which mandate considerable updates to the SVG code, currently lie outside the proficiency range of these models.

In summary, while LLMs do present limitations, it offers promising initial results for the integration of LLMs and SVG for visual tasks. Addressing these limitations could lead to more powerful image representation algorithms and pave the way for more versatile and comprehensive artificial intelligence systems.

## 5 CONCLUSION

This paper explored the possibility of enabling large language models (LLMs) to "see" and process images through the Scalable Vector Graphics (SVG) format. By converting raster images into SVG representations and leveraging XML-based textual descriptions, we showed that LLMs have some ability understand and manipulate images.

We studied LLMs' capabilities across various visual reasoning, recognition, and generative tasks, revealing the underlying shape-color disentanglement nature of SVG. Through these experiments, we showed that SVG representation shows better performance compared to the closed-set trained model, and could continue refine the outcome with chat-based feedback.

This research can open the door to new opportunities in the realm of computer vision by integrating the powerful capabilities of LLMs with SVG format. We believe that our work provides an initial exploratory step for future research in the integration of LLMs and SVG for the development of advanced image representation formats and more complex vision tasks. As we continue to explore the potential of large language models on visual input, this approach could inspire further progress in the understanding of visual data with multi-modal fusion approaches.

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

APPENDIX

## A  EXPERIMENT DETAILS

### A.1  DATASET

**Human Designed SVG Dataset** We collect a dataset from the public collection of SVG images.[1] Specifically, we collect the digits and icons to demonstrate image recognition and generation capabilities. Examples are shown in Figure 6 (a) and (b).

**Convert Raster Images to SVG** 1) Directly convert using curve tracing. Given the rich set of natural images in raster format, we utilize the curve tracing algorithm to convert RGB images into the SVG format.[2] Specifically, we convert MNIST (LeCun et al., 2010) to SVG format using this approach, shown in Figure 6 (c).

---

[1]`https://www.svgrepo.com/`,     `https://www.kaggle.com/datasets/victorcondino/svgicons`

[2]`https://github.com/visioncortex/vtracer`

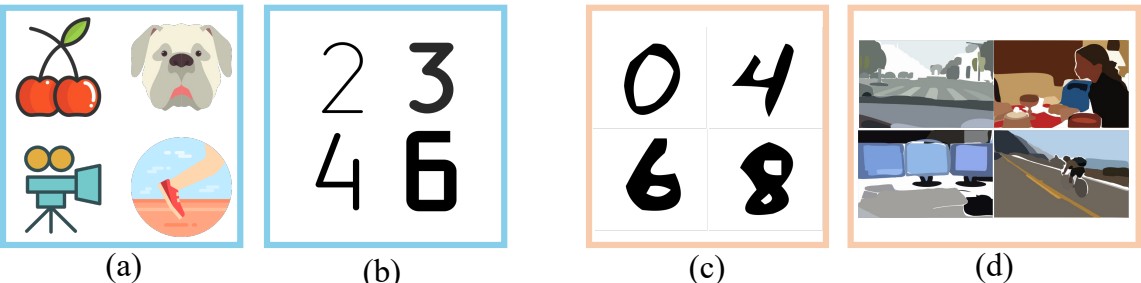

Figure 6: Visualization of our datasets. (a) and (b) are human-designed SVG vectors and icons. (c) and (d) are converted from raster images. Specifically, (c) is generated using curve tracing from MNIST (LeCun et al., 2010), while (d) is generated using SAM (Kirillov et al., 2023) and curve tracing sequentially.

### A.2 RASTER IMAGES TO SVG CONVERSION

One of the most fundamental pieces of information for visual perception is object shape. Our method can be conceptualized as selectively diminishing details from an image, prioritizing the extraction of less significant shapes. This guided process of reduction offers a quantitative way to manage the amount of visual data present within an image. Within this framework, we perceive segmentation as an example of extreme simplification, whereas vectorization serves as a more moderate form of the same. Here we introduce how we use such two approaches to convert the raster images to SVG.

**Image Vectorization.** The vector tracing algorithm operates in a sequential three-step process. Initially, pixels are transformed into a defined path. Subsequently, this path is condensed into a simplified polygonal representation. Lastly, the polygon is refined and approximated using a curve-fitting (tracing) technique, which enhances its smoothness.

There are several online tools to convert the raster images (jpg and png) into vector graphics (SVG), such as Adobe Illustrator (Adobe Inc., 2019), Inkscape (Inkscape Project, 2020), and VTracer (VTracer, 2020). We experiment with all of them and found that VTracer leads to the best balance between SVG complexity (code length) and rich semantic representation.

In MNIST (LeCun et al., 2010), we use the default hyperparameters during conversion. Specifically, we (i) first binarize the MNIST pixel value from the continuous range [0, 255] to the binary set $\{0, 255\}$ using the threshold 127.5, (ii) set the foreground to black, and the background to white, and (iii) apply the vector tracing algorithm VTracer.

**Segmentation Prior.** As mentioned earlier, segmentation can provide a strong prior for object shape. We want a generalist model that can segment any image, i.e., not trained and thus biased towards a certain dataset. The Segment Anything (SA) (Kirillov et al., 2023) project introduces such an image segmentation model, the Segment Anything Model (SAM), and a large-scale dataset, SA-1B, with the aim of achieving powerful generalization and zero-shot transfer across diverse segmentation tasks, demonstrating competitive results often surpassing prior fully supervised methods. We use the default hyper-parameters of SAM to segment the whole image into a set of masks without class labels, where the color of each mask is represented by the average value of the pixels within the mask. Specifically, we sample 32 query points per side (1024 points overall) to generate the image mask. Then we select the top 20 masks with the highest area as the final representation for an image.

We then use VTracer to transform the mask into SVG format. Note that, to reduce the final SVG, we adjust several settings: we set the number of significant bits to use in an RGB channel to 0; we set the minimum angle displacement degree to splice a spline to 90; we set the color difference between gradient layers to be 35; we consider a corner to have a minimum momentary angle of 0 degrees; we discard patches smaller than 16 pixels in size; and we perform iterative subdivide smoothing until all segments are shorter than 10 pixels.

### A.3 FINE-TUNING DATASET FOR VICUNA

We use the same JSON format in Vicuna (Vicuna, 2023) to construct the fine-tuning dataset. We use all the training samples in MNIST, translating to 60,000 SVG images. For each sample, we construct one round of conversation: (i) From human: ``Which digit does the following SVG reflect? <SVG code here>'', and (ii) From GPT: ``<label>''. Here <label> denotes the digit label, which ranges from 0 to 9. Then we use this dataset to fine-tune Vicuna using the default hyper-parameters [3] for 3 epochs.

### A.4 PROMPT ENGINEERING

In this section, we provide the details of prompt engineering for each task. The prompt is designed to figure out the common transformation in the SVG example pairs first (each example pair consists of a query and a key) and then transform the query into the corresponding key SVG by following the discovered common transformation.

**In-context Image Classification.** In this task, in-context examples are aimed to provide more context information using several image-label pairs, thus facilitating the final classification. The specific prompt utilized for this purpose using 3 in-context examples is detailed below: ``Instruction: please predict the digit number for each of the following SVG images. Please think step by step, and closely look at the key identifying image characteristics. Please just tell me the image class, no other information is needed. Q: What digit does this SVG image represent? <SVG code here> A: This SVG image represents digit <label> Q: What digit does this SVG image represent? <SVG code here> A: This SVG image represents digit <label> Q: What digit does this SVG image represent? <SVG code here> A: This SVG image represents digit <label> Q: What digit does this SVG image represent? <SVG code here> A: This SVG image represents digit .

**Synthetic Data Study:** In this task, the objective is to conduct an analytical evaluation of the provided two example pairs, examining changes that occur in aspects such as color, shape, and size. The insight gathered from this analysis will then be used to adapt the given query into its corresponding key. The specific prompt utilized for this purpose is detailed below: ``Please perform the following task carefully. In this task, you will be shown two examples of Scalable Vector Graphics (SVG) code pairs. Each pair consists of a query and key pair, where the query describes the SVG code of the original image, and the key describes the SVG code of the transformed image. Each will be named ``Example Query #" and ``Example Key #" respectively. Your first task is to figure out the common transformation in the two examples. The transformation can consist of color, shape, size, or any combination thereof. Then, given a new test query SVG code (named

---

[3] https://github.com/lm-sys/FastChat

```
\Test Query"), your second task is to transform that query into the
corresponding key SVG code (named \Test Key"), following the common
transformation that you discovered in the two example pairs.  Here
are the two example query and key pairs:  Example Query 1:  <SVG code
here>; Example Key 1:<SVG code here>; Example Query 2:<SVG code here>;
Example Key 2:<SVG code here>; Here are the test query and key pair:
Test Query:<SVG code here>; Test Key:''
```

**Content Extrapolation:** In this task, LLMs are presented with SVG code pairs, each containing a query-key set that depicts numbers. The key introduces a consistent mathematical operation (addition, subtraction, multiplication, or division) to the query number. The tasks are to identify this operation in the examples and apply it to new test queries to generate corresponding test keys. To facilitate a more comprehensive understanding of SVG number codes for the LLM, we initially present the SVG codes for numbers 0 through 9 to the LLM prior to posing specific queries. The specific prompt utilized for this purpose is detailed below: ``Please perform the following task

```
carefully.  In this task, you will be shown two examples of Scalable
Vector Graphics (SVG) code pairs.  Each pair consists of a query and
key pair, where the query describes an SVG code of an integer number,
and the key describes the SVG code of another integer number with an
introduced mathematical operation.  Each will be named \Example Query
#" and \Example Key #" respectively.  In addition to the example pairs,
you will be shown a new test query SVG code (named \Test Query").  Your
first task is to identify which number each example query, example key,
and test query represents.  Your second task is to figure out all the
possible mathematical operations that are held for all given example
pairs.  The operation could be add, subtract, multiply, and divide (the
subtract or multiply factor could be a fraction).  Then, according to
the numbers of example pairs and test query you identified, your third
task is to predict the corresponding test key number (named \Test Key"),
following all the mathematical operations that you discovered in the
given example pairs.  Finally, you need to generate the corresponding
SVG code of the test key number.  Here are the two example query and
key pairs:  Example Query 1:  <SVG code here>; Example Key 1:<SVG code
here>; Example Query 2:<SVG code here>; Example Key 2:<SVG code here>;
Here are the test query and key pair:  Test Query:  <SVG code here>;
Test Key:  (Note:  think about four operations one by one, and the
operation should be consistent for all given example pairs)''
```

## B  VISUAL REASONING RESULTS OF LLAVA

Here we evaluate the recent multi-modal large language model, LLaVa (Liu et al., 2023). As the result in Table 4 shows, LLaVa struggles at this fundamental reasoning task. Besides,we observe that LLaVa frequently defaults to 'yes' for yes/no queries and often resorts to random guessing for counting tasks. This behavior underscores the limitations of current large multimodal models in structured and sophisticated reasoning.

## C  QUALATATIVE CHAT RESULTS

**Image Recognition and Manipulation.** In this section, we provide examples for chat-based image recognition and manipulation using GPT4 (OpenAI, 2023b). The qualitative results are shown

Table 4: Catogori-wise accuracy on the Sort-of-Clevr dataset.

| Question type | GPT4-brief | GPT-CoT | LLaVa | CNN+MLP | Relation Networks |
|---|---|---|---|---|---|
| Unary | 0.50 | 0.90 | 0.60 | 0.65 | 0.89 |
| Binary | 0.90 | 0.95 | 0.60 | 0.75 | 0.80 |
| Ternary | 0.10 | 0.88 | 0.10 | 0.55 | 0.55 |
| Average | 0.50 | 0.89 | 0.43 | 0.65 | 0.75 |

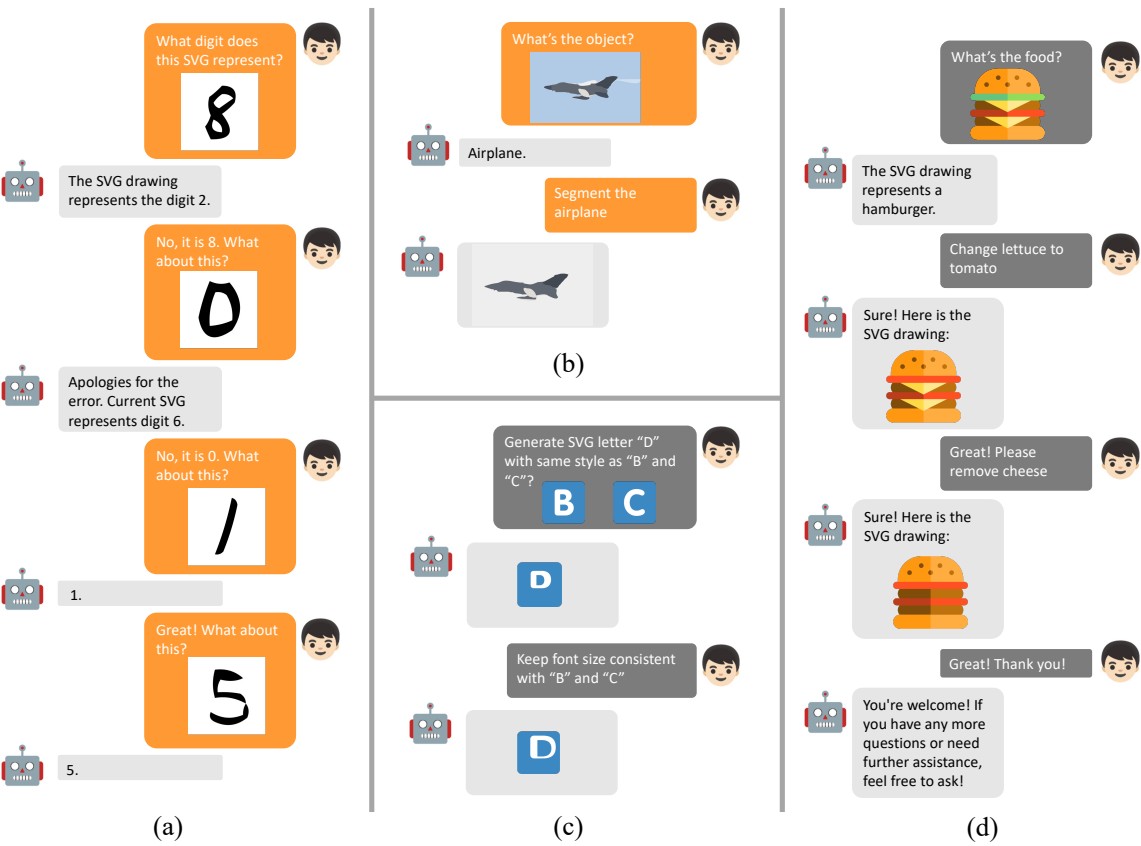

Figure 7: In-context learning and image generation capabilities of SVG with LLMs. (a) With human feedback, LLM gradually performs better on digit classification. (b) LLM powers SVG with the capability of image recognition and referring segmentation. (c) With human feedback, the content generation performance becomes better. (d) LLM can recognize and manipulate specific parts of the hamburger, such as removing or replacing them.

in Figure 7 and 8: (a) SVG representation empowers robust in-context digit recognition capability given different background and foreground colors, (b) GPT4 can recognize and depict the details of a dog with the prompt: `"a stylized bear or a similar mammal with a round face and ears."` Furthermore, GPT-4 can identify the location of the dog's left eye and remove it. (c) GPT4 is capable of recognizing a natural image from the CIFAR-10 dataset.

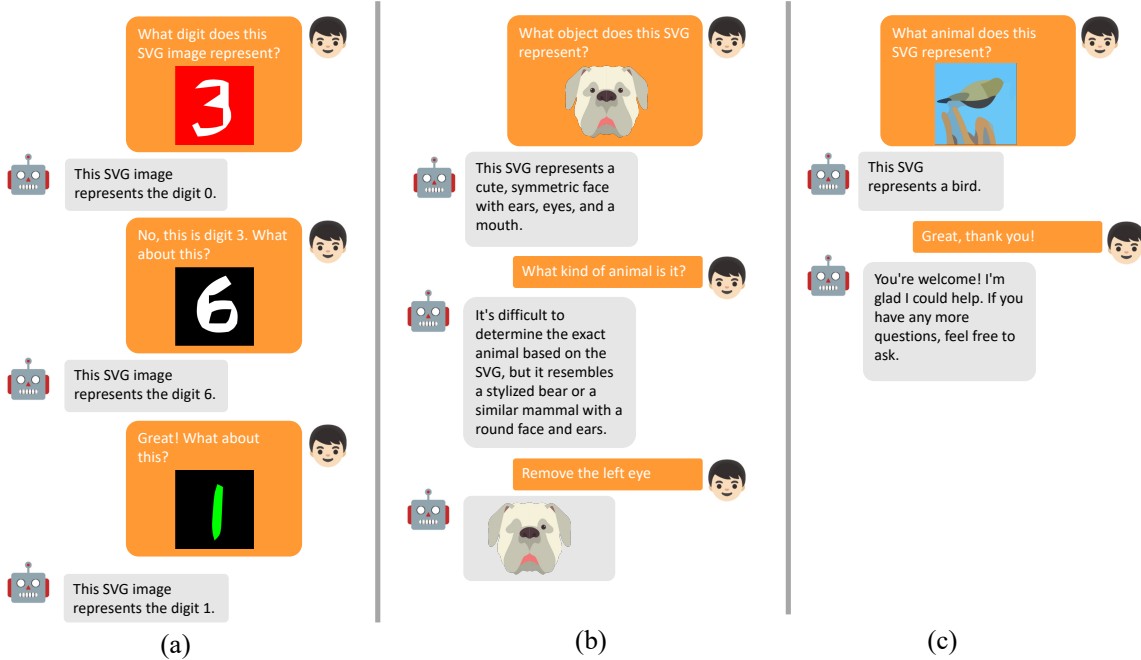

(a)  (b)  (c)

Figure 8: More qualitative results of chat-based image recognition and manipulation. (a) In-context digit recognition in Colored-MNIST-(B). (b) GPT can explain and manipulate the dog SVG image. (c) GPT4 can also recognize the bird from a CIFAR-10 example.

**Referring Segmentation**  The objective of the task is to label pixels in an image or video that correspond to an object instance referred by a linguistic expression. SVG representation has two advantages. First, language instruction is naturally embedded within the prompt, thus a separate design of the image segmentation model is not needed. Second, a large corpus of text and programming languages including XML are seen during pretraining, benefiting the vision-language understanding ability.

SVG is typically composed of several colored polygons, where each of them can correspond to a part of the object. Therefore, we can use the referring segmentation instructions to guide the LLM in finding the corresponding SVG code. Shown in Figure 7 (b) and (d), LLM can localize the object decently well. In (b), the majority of the airplane is selected as foreground, while in (d), not only is the lettuce recognized, but also the two pieces of cheese are localized and subsequently removed.

**Style Extrapolation:**  LLMs are provided with five example pairs and are tasked with deciphering the stylistic attributes inherent in these examples.  Following this, a new test query is presented to the LLMs.  Their objective is to modify this query into the corresponding key, ensuring that it aligns with the same stylistic principles showcased in the example pairs.  The qualitative results

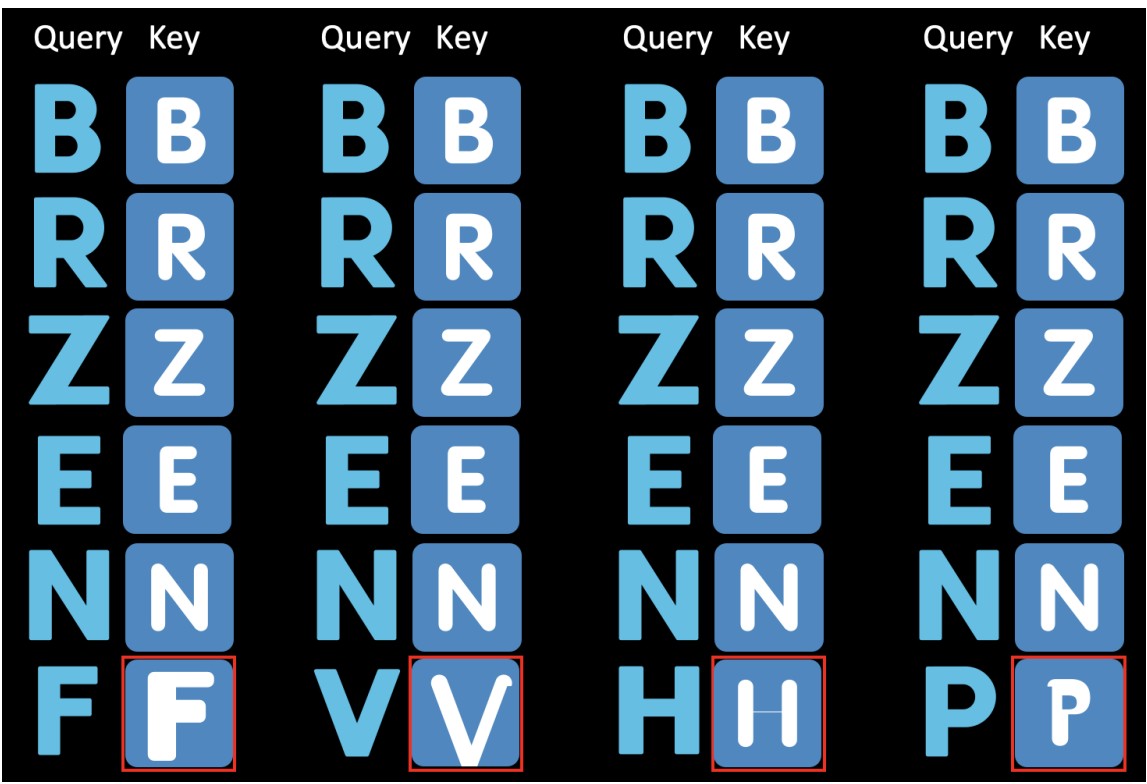

Figure 9: More qualitative results of style extrapolation. The generation results of our method are annotated with a red square.

are shown in figure 9. The specific prompt utilized for this purpose is detailed below:``Please perform the following task carefully. In this task, you will be shown five examples of Scalable Vector Graphics (SVG) code pairs. Each pair consists of a query and key pair (both are English letter), where the query describes the SVG code of the original image, and the key describes the SVG code of the transformed image. Each will be named \Example Query #" and \Example Key #" respectively. Your first task is to figure out the common transformation in the five examples. The transformation can consist of color, shape, size, style, font, and background changes, or any combination thereof. Even though you cannot see the images, and only their SVG codes, you need to discover the transformations that are happening at the image level and not just at the code level. Be detailed, and try to discover every change, and the most important change is that the paths in the SVG code between each query and key is different due to the common transformation but the shapes of the letters that query and key are representing remain the same. Then, given a new test query SVG code (named \Test Query"), your second task is to transform that query into the corresponding key SVG code (named \Test Key"), following the common transformation that you discovered in the five example pairs. To help you better understand the transformation, I will also inform you of what letter each query and key represent. You need to find the shape of each query and key by analyzing their path. Here are the five example query and key pairs: Example Query 1 (letter B):; Example Key 1 (letter B):<SVG code here>; Example Query 2 (letter R):<SVG code here>; Example Key 2 (letter R):<SVG code here>; Example Query 3 (letter Z):<SVG code here>; Example Key 3 (letter Z):<SVG code here>; Example Query 4 (letter E):<SVG code here>; Example Key 4 (letter E):<SVG code here>; Example Query 5 (letter N):<SVG code here>; Example Key 5 (letter N):<SVG code here>; Here is the test query and key pair: Test Query (letter #):; Test Key: ''

