# OpenReview forum: "Delving into LLMs’ visual understanding ability using SVG to bridge image and text"
_ICLR.cc/2024/Conference — Submitted to ICLR 2024_

### Official Review · Reviewer_oVfB · 2023-10-22

**Soundness:** 3 good
**Presentation:** 3 good
**Contribution:** 3 good
**Rating:** 6
**Confidence:** 4

**Summary:**

This paper explores the capability of pre-trained LLM in understanding images through converting images into readable text via SVG. The authors evaluate both discriminative and generative visual understanding tasks including image classification, image generation and editing.

**Strengths:**

1. The study of whether LLMs, which have never seen visual data, can understand and reason about images is interesting and novel.
2. This paper utilizes SVG to convert images into structured XML codes, which serves as a bridge to apply the analytical strengths of LLMs to the visual domain.
3. This paper evaluates both discriminative and generative visual understanding tasks of LLMs, and draws some interesting conclusions, which can be inspiring.

**Weaknesses:**

1. For the evaluation of Multimodal LLMs, the authors only experiment on LLaVa and conclude that "This behavior underscores the limitations of current large multimodal models in structured and sophisticated reasoning.", which is not rigorous. The authors should evaluate more Multimodal LLMs such as BLIP-2, InstructBLIP, MiniGPT-4,mPLUG-Owl, etc.
2. The data for the evaluation of both the discriminative and generative visual understanding tasks are relatively simple. How does the LLM perform with SVG representations on more complicated images? Since the authors also mention that SVG is not effective in handing photographic content, using SVG with LLMs to tackle visual tasks may not be effective enough.

**Questions:**

See Weaknesses.

**Details Of Ethics Concerns:**

In my opinion, no ethics review is needed.

---

> ### Author Response · Authors · 2023-11-20
> **Response to Reviewer oVfB**
>
> Dear Reviewer oVfB,
>
>
> We are grateful for your acknowledgment of our study’s novelty, and comprehensive evaluation of both discriminative and generative visual understanding of LLMs. For your questions—we'll make revisions accordingly.
>
> > “The authors should evaluate more Multimodal LLMs such as BLIP-2, InstructBLIP, MiniGPT-4, mPLUG-Owl, etc”
>
> We appreciate your suggestion. We have expanded our comparison in the table to include additional Multimodal LLMs. The updated results indicate that combining SVG with LLMs outperforms other models like BLIP-2, InstructBLIP, MiniGPT-4, and mPLUG-Owl in handling complex, structured reasoning tasks.
>
> | Question Type | GPT-CoT | LLaVa | CNN+MLP | Relation Networks | InstructBLIP (13b) | BLIP2 (Flan T5-xxl) | mPLUG_owl | MiniGPT4 (13B) |
> |---------------|---------|-------|---------|-------------------|--------------------|---------------------|-----------|----------------|
> | Format        | SVG     | PNG   | PNG     | PNG               | PNG                | PNG                 | PNG       | PNG            |
> | Unary         | 0.90    | 0.60  | 0.65    | 0.89              | 0.53               | 0.50                | 0.38      | 0.53           |
> | Binary        | 0.95    | 0.60  | 0.75    | 0.80              | 0.53               | 0.53                | 0.63      | 0.55           |
> | Ternary       | 0.88    | 0.10  | 0.55    | 0.55              | 0.10               | 0.30                | 0.30      | 0.30           |
> | Average       | 0.89    | 0.43  | 0.65    | 0.75              | 0.38               | 0.44                | 0.43      | 0.46           |
>
> _Caption: Performance comparison across different models on the Sort-of-Clever Dataset._
>
> > “How does the LLM perform with SVG representations on more complicated images?”
>
> Thank you for raising this concern. We agree that the current approach of integrating SVG with LLMs for complex visual tasks may fall short, primarily due to the loss of fine details in converting and the lengthy sequences of encoding extensive details. To tackle this, a potential solution is to leverage large vision models to extract the essence of images and convert that essence into the SVG format for LLM processing. An example is utilizing SAM [1] for object mask segmentation, which is then translated to SVG for LLMs, as depicted in Figure 6 (d). We then show a recognition result in Figure 7 (b).
>
> [1] Kirillov, Alexander, Eric Mintun, Nikhila Ravi, Hanzi Mao, Chloe Rolland, Laura Gustafson, Tete Xiao et al. "Segment anything." ICCV (2023).

---

> > ### Comment · Reviewer_oVfB · 2023-11-23
> >
> > I have read the reponse and keep my rating.

---

> > > ### Author Response · Authors · 2023-11-23
> > > **Thanks for your reply!**
> > >
> > > Dear Reviewer oVfB,
> > >
> > > We sincerely appreciate your insightful feedback and constructive suggestions on our paper. Your thorough review has contributed to enhancing the quality and rigor of our work. Thank you for your valuable input and the time you have dedicated to our submission.

---

> ### Author Response · Authors · 2023-11-22
> **Looking forward to your reply!**
>
> Dear reviewer oVfB,
>
> Thank you for reviewing our work to enhance the quality of the paper!
> Has our rebuttal adequately addressed your concerns? If you still have any issues with our rebuttal or if there are any new concerns, we are more than willing to continue the discussion with you.

---

### Official Review · Reviewer_Vo5P · 2023-10-31

**Soundness:** 3 good
**Presentation:** 3 good
**Contribution:** 2 fair
**Rating:** 6
**Confidence:** 3

**Summary:**

The paper explores the potential of Large Language Models (LLMs) to understand and process images by converting them into Scalable Vector Graphics (SVG), which can be represented as an XML-based textual description. The LLM's abilities are tested across three vision tasks: visual reasoning, image classification under distribution shifts, and generating new images based on visual prompts. The results show that the LLM has a reasonable ability to perform these tasks, especially compared with some expert models.

**Strengths:**

1. The use of SVGs to turn images into text is an innovative way of exploiting the potential of LLMs for image processing tasks.
2. The paper conducts both qualitative and quantitative evaluations over a variety of visual understanding tasks providing deep insights into the LLMs' image understanding capabilities. The results indicate that LLMs can perform well even when there are distribution shifts in visual data, demonstrating robustness. The demonstration of LLMs' ability to generate and edit images based on chat-based feedback is particularly promising and marks a significant advancement.

**Weaknesses:**

1. The SVG representation may not capture all the nuances of an image. It is yet unclear how well this method would generalize to more complex images or visual tasks.
2. The issue raised in the introduction about whether LLMs can learn world models without grounding in physical interaction and visual perception remains unaddressed. A comparison of LLMs' performance using SVGs with traditional vision-based models could have provided more context about the relative effectiveness of this approach.

**Questions:**

1. How well do SVG representations capture complex images? Could you provide instances or examples where this approach has limitations?
2. Whether the format, sequence, or other characteristics of textual descriptions in Scalable Vector Graphics (SVG) will affect their subsequent task performance.
3. What could be the potential impact if a minor percentage of errors into the representation of SVGs is introduced?

---

> ### Author Response · Authors · 2023-11-21
> **Response to Reviewer Vo5P (Part I)**
>
> Dear Reviewer Vo5P,
>
> We are grateful for your acknowledgment of our study’s novelty, comprehensive evaluation, and promising results. For your questions; we'll make revisions accordingly.
>
> > "How well do SVG representations capture complex images? Could you provide instances or examples where this approach has limitations?"
>
> Thank you for pointing out this concern. We acknowledge the limitations of using SVG with LLMs for complex visual tasks, due to the loss of fine-grained details during conversion and the challenge of managing prohibitively long sequences when encoding detailed photographic content. To tackle this, a potential solution is to leverage large vision models to extract the essence of images and convert that essence into the SVG format for LLM processing. An example is utilizing SAM [1] for object contour segmentation, which is then translated to SVG for LLMs, as depicted in Figure 6 (d). We then show a recognition result in Figure 7 (b).
>
> > "A comparison of LLMs' performance using SVGs with traditional vision-based models could have provided more context about the relative effectiveness of this approach."
>
> Thank you for your suggestion. We'd like to clarify that in our paper, we have compared the performance of LLMs using SVGs with traditional vision-based models. This includes:
>
> - Visual reasoning tasks in Table 1, where LLMs with SVGs outperform SoTA vision-based models in both in-distribution and out-of-distribution scenarios.
>
> - Image classification tasks in Table 2, showing LLMs with SVGs again surpassing SoTA vision-based models under similar conditions.
>
> - Visual prompting in Table 3, where LLMs with SVGs demonstrate superior performance compared to SoTA vision-based models.
>
>
> [1] Kirillov, Alexander, Eric Mintun, Nikhila Ravi, Hanzi Mao, Chloe Rolland, Laura Gustafson, Tete Xiao et al. "Segment anything." ICCV (2023).

---

> ### Author Response · Authors · 2023-11-21
> **Response to Reviewer Vo5P (Part II)**
>
> > "Whether the format, sequence, or other characteristics of textual descriptions in Scalable Vector Graphics (SVG) will affect their subsequent task performance"
>
> and
>
> > "What could be the potential impact if a minor percentage of errors into the representation of SVGs is introduced?"
>
> Thanks for both of your suggestions. To rigorously evaluate the robustness of LLMs against variations in SVG data, we conducted three distinct experiments, where we: (i) shuffle the order of paths, (ii) randomize path coordinate replacement, and (iii) randomize string replacement. Each experiment was designed to mimic real scenarios of imperfections in SVG data, providing insights into using SVG with LLMs under challenging conditions.
>
> __(i) Path Shuffle Experiment__ SVG data consists of multiple path elements, each representing an object or line in the image. In this experiment, we tested the model's ability to interpret hand-drawn SVG data from the MNIST dataset when the sequence of path elements was shuffled. Note that, by the very nature of SVG data, this alteration will __not__ change the final image; and hence, it is important to test if the LLM can remain invariant to this change, even though we have not explicitly provided it with this knowledge.
> The goal is to assess the model's ability to correctly interpret the digit, regardless of the order of path elements. The results are summarized in the table below. Results indicate that SVG is robust to path shuffle at test time, maintaining a comparable performance compared to the SVG representation with the original path ordering.
>
>
> | Condition       | Accuracy (%) |
> |-----------------|--------------|
> | Without Shuffle | 99.10        |
> | With Shuffle    | 98.74        |
>
> _Caption: Model accuracy for the path shuffle experiment._
>
> __(ii) Random Path Coordinate Replacement__
> We follow the reviewer's suggestion and introduce a subtle form of noise by randomly altering the coordinates in the SVG path data. Each numerical value within the path commands is randomly translated within a specific range $k$. For example, for $k=1$,   `... <path d='M0 0 C18 0 17...'>` might become `... <path d='M1 0 C18 1 18...'>`. We test several variations: (i) a minor adjustment within a 1/28th range (reflecting the 28x28 resolution of MNIST) and (ii) a more significant alteration within a 5/28th range. This simulates real-world scenarios of minor inaccuracies in SVG data, such as those resulting from conversion errors or imprecise digitization.
> Table shown below presents the accuracy under different noise scales. Results indicate data SVG representation is decently robust to the perturbation of the coordinate values.
>
> | Noise Scale | Accuracy (%) |
> |-------------|--------------|
> | 0/28        | 99.10        |
> | 1/28        | 98.97        |
> | 2/28        | 97.91        |
> | 5/28        | 87.56        |
>
> _Caption: Model accuracy under different levels of coordinate noise._
>
> __(iii) Random String Replacement__
> Finally, we design the most aggressive test of robustness, where we replace random characters in the SVG strings with any English alphabet letter, digit, or special symbol, regardless of whether they are numerical values or part of SVG commands (like `transform='translate(0,0)'`). The experiment is conducted with varying probabilities for character replacement, as shown below.
> Surprisingly, even after replacing 20\% of a string with random characters, SVG with LLMs maintains a high accuracy rate of 90.79\%. This suggests that SVG with LLMs can handle a wide range of perturbations in SVG data.
>
>
> The following table presents the accuracy with varying probabilities of random string replacement:
>
> | Replacement Probability (%) | Accuracy (%) |
> |-----------------------------|--------------|
> | 0                           | 99.10        |
> | 1                           | 99.06        |
> | 5                           | 98.40        |
> | 10                          | 97.40        |
> | 20                          | 90.79        |
> | 50                          | 39.38        |
>
> _Caption: Accuracy with different probabilities of random string replacement._
>
> In conclusion, the results from these experiments demonstrate the robustness of using SVG data with LLMs for visual understanding. Despite the introduction of perturbations, using SVG data with LLMs can perform well under challenging conditions.

---

> ### Author Response · Authors · 2023-11-22
> **Looking forward to your reply!**
>
> Dear reviewer Vo5P,
>
> Thank you for reviewing our work to enhance the quality of the paper!
> Has our rebuttal adequately addressed your concerns? If you still have any issues with our rebuttal or if there are any new concerns, we are more than willing to continue the discussion with you.

---

> > ### Comment · Reviewer_Vo5P · 2023-11-23
> >
> > Thank you for your diligent response in addressing the issues I previously raised. Your efforts have not gone unnoticed and I appreciate the time you've spent on this task. As a result of these changes, I've decided to adjust my rating to a 6.

---

> > > ### Author Response · Authors · 2023-11-23
> > > **Thanks for your reply!**
> > >
> > > Dear Reviewer Vo5P,
> > >
> > > Thank you for your thoughtful and encouraging feedback!
> > > We are truly grateful for your constructive comments and the positive impact they have had on our work. Your guidance has been instrumental in helping us explore this area more deeply and rigorously. We sincerely appreciate the time and effort you have devoted to reviewing our submission and for helping us enhance the quality of our research.

---

### Official Review · Reviewer_uhQW · 2023-11-07

**Soundness:** 2 fair
**Presentation:** 3 good
**Contribution:** 2 fair
**Rating:** 5
**Confidence:** 4

**Summary:**

They use Scalable Vector Graphics for using LLM for image understanding. The authors provide various experiments, including both discriminative and generative visual understanding tasks.

**Strengths:**

1. **Experimental Rigor:** The authors present a comprehensive suite of experiments in the domain of visual understanding and reasoning that effectively leverages the combination of Scalable Vector Graphics (SVG) with large language models (LLMs). The breadth and depth of the experimental design are commendable and provide valuable insights into the capabilities of LLMs in processing vector-based graphic representations.

2. **Robustness to Distribution Shifts:** A significant strength of the paper is the demonstration of the robustness of the LLM + SVG approach under conditions of distributional shifts.

3. **Clarity and Coverage of Related Works:** The paper is well-articulated, presenting its concepts and findings in a manner that is accessible to readers. Moreover, the authors have done a thorough job in situating their work within the context of existing literature. They have effectively covered pertinent related works.

**Weaknesses:**

1. **Component Originality:** While the experiments are detailed, the novelty of the individual components used within the work is unclear. The application of Scalable Vector Graphics (SVG), Large Language Models (LLMs), the dataset selection, the tasks, and the chosen evaluation metrics all appear to be repurposed from existing literature without significant innovation or new application. For a stronger contribution, the authors could benefit from integrating at least one novel element or a unique combination of these elements that distinguishes this work from prior studies.


2. **Source of Performance Discrepancy:** The comparative results between LLM+SVG and CNN+PNG in the first experiment are intriguing but lack a clear explanation. The performance gap could be attributed to differences in input representation or model architecture, among other factors. An in-depth ablation study could help isolate the impact of each component. Furthermore, the disparity in model sizes (e.g., GPT-4's size relative to that of a CNN) is a confounding variable that merits consideration. To convincingly argue the advantages of using SVG for image representation in LLMs, the authors should compare SVG with alternative representations using the same underlying model architecture, such as GPT-4.


3. **Dataset and Task Relevance:** The chosen dataset and task do not seem to reflect complex real-world scenarios and may inherently favor SVG representations due to their structured nature. This could introduce a bias towards the advantages of SVG in image understanding tasks that focus on structural rather than fine-grained details. The authors would enhance the robustness of their findings by including a broader range of tasks with varying degrees of complexity and realism to demonstrate the efficacy of SVG representations across different contexts.


4. **Methodological Limitations:** The exploration of image understanding through LLMs is indeed crucial; however, the analysis presented in this work is constrained by its reliance on existing methodologies. The absence of innovative input representations, model adaptations, or new evaluation benchmarks suggests a missed opportunity for advancing the field. To forge a more impactful contribution, the authors should strive to develop and introduce novel methodologies or significantly adapt existing ones to the specific challenges of image understanding through LLMs.

**Questions:**

Refer to part of Weaknesses.

---

> ### Author Response · Authors · 2023-11-21
> **Response to Reviewer uhQW (Part I)**
>
> Dear Reviewer uhQW,
>
>
>
> We are grateful for your acknowledgment of our study’s comprehensive evaluation, and demonstration of the robustness of the LLM + SVG under distribution shift. Below, we respond to your specific questions.
>
>
> > “While the experiments are detailed, the novelty of the individual components used within the work is unclear”
>
>
> We wish to clarify that our study primarily aims to provide a thorough analysis of the strengths and weaknesses of employing LLMs to understand and reason about images without visual data. Below, we detail the novelty of each component in our study.
>
>
> - Comprehensive study of LLMs across various visual tasks using SVG format, including visual reasoning, image classification, and image generation.
> - Investigation into LLMs' performance in visual data using SVG format under conditions of distribution shifts, expanding our understanding of their robustness.
> - Analysis of LLMs' generative capabilities in image generation and editing based on interactive feedback within the SVG format.
>
> > “The comparative results between LLM+SVG and CNN+PNG in the first experiment are intriguing but lack a clear explanation. ... isolate the impact of each component.”
>
> Thank you for your suggestion. Following your advice, we conducted additional experiments to assess the impact of different input formats using the same model, as shown in the table below. Specifically, we compare the SVG input format with GPT4 against PNG input format with GPT4V (likely larger than GPT4 due to its visual encoding network component) for a balanced evaluation. The results indicate that using SVG with GPT4 significantly outperforms using PNG with GPT4V, in complex, structured reasoning tasks.
>
>
>
> | Question Type | GPT-CoT | GPT-4V | LLaVa | CNN+MLP | Relation Networks | InstructBLIP (13b) | BLIP2 (Flan T5-xxl) | mPLUG_owl | MiniGPT4 (13B) |
> |---------------|---------|--------|-------|---------|-------------------|--------------------|---------------------|-----------|----------------|
> | Format        | SVG     | PNG    | PNG   | PNG     | PNG               | PNG                | PNG                 | PNG       | PNG            |
> | Unary         | 0.90    | 0.75   | 0.60  | 0.65    | 0.89              | 0.53               | 0.50                | 0.38      | 0.53           |
> | Binary        | 0.95    | 0.74   | 0.60  | 0.75    | 0.80              | 0.53               | 0.53                | 0.63      | 0.55           |
> | Ternary       | 0.88    | 0.28   | 0.10  | 0.55    | 0.55              | 0.10               | 0.30                | 0.30      | 0.30           |
> | Average       | 0.89    | 0.59   | 0.43  | 0.65    | 0.75              | 0.38               | 0.44                | 0.43      | 0.46           |
>
> _Caption: Performance comparison across different models on the Sort-of-Clever Dataset._
>
>
>
>
>
> Additionally, to ensure a fair comparison, we compared the performance of both SVG and PNG inputs using the same Vicuna model for image classification in the table below
> As shown in the following figure, under the same model architecture, results indicate that SVG is more robust than rasterized representation.
>
>
>
>
>
> | Method                 | ConvNeXt (fine-tuning) | Vicuna (fine-tuning) | Vicuna (fine-tuning) |
> |------------------------|------------------------|----------------------|----------------------|
> | Image Format           | PNG                    | SVG                  | PNG                  |
> | MNIST                  | 99.5%                  | 99.1%                | 99.4%                |
> | CMNIST-(A)             | 79.5%                  | 95.7%                | 42.9%                |
> | CMNIST-(B)             | 32.6%                  | 92.9%                | 24.8%                |
>
> _Caption: Performance comparison across different models on the MNIST Dataset._
>
>
>
>
> > “The chosen dataset and task do not seem to reflect complex real-world scenarios and may inherently favor SVG representations due to their structured nature.”
>
>
>
> We thank the reviewer for raising and highlighting an important point. First, we would like to re-emphasize the main goal of our work: to see how much knowledge LLMs already have in understanding visual data, even though one might not naturally assume them to be suited for any vision task. Therefore, any experiments that we conduct (e.g., Table 1) should be seen as a way to study that question, and _not necessarily_ as a way to demonstrate that LLMs are fundamentally superior to CNN/transformers. And since, to the best of our knowledge, our work is the first to thoroughly study this hidden capability of LLMs, we have chosen the simpler settings. Our hope, however, is that our work can motivate investigations into other, more complex, vision tasks that can be solved with the help of an LLM.

---

> ### Author Response · Authors · 2023-11-21
> **Response to Reviewer uhQW (Part II)**
>
> > “The analysis presented in this work is constrained by its reliance on existing methodologies. The absence of innovative input representations, model adaptations, or new evaluation benchmarks suggests a missed opportunity for advancing the field.”
>
>
> We recognize that while our study relies on established techniques, these techniques are sufficiently effective for practical use in simple scenarios. We agree that addressing complex real-world images may necessitate the development of new methods. We are grateful for being the pioneers to start the investigation of LLMs for understanding and reasoning about images without relying on visual data, and we are committed to advancing our research to tackle real-world challenges more effectively.

---

> ### Author Response · Authors · 2023-11-22
> **Looking forward to your reply!**
>
> Dear reviewer uhQW,
>
> Thank you for reviewing our work to enhance the quality of the paper!
> Has our rebuttal adequately addressed your concerns? If you still have any issues with our rebuttal or if there are any new concerns, we are more than willing to continue the discussion with you.

---

> > ### Comment · Reviewer_uhQW · 2023-11-23
> > **Thank you for rebuttal**
> >
> > Thanks for your detailed answer and additional experiments.
> > My concern about the impact of each component is addressed.
> > However, I still think the tasks and benchmarks are biased to the SVG + LLM setting.
> > It would be great if you
> > 1) introduce a new dataset or benchmark for evaluating LLMs’ visual understanding ability, OR
> > 2) utilize real-world tasks or benchmarks to evaluate LLMs’ visual understanding ability.
> >
> > So the effort of the work is clear, but its experiments (task, dataset, benchmark) are not suitable for evaluating LLMs’ visual understanding ability.

---

> ### Author Response · Authors · 2023-11-23
> **Thanks for your reply!**
>
> Glad that we have addressed your aforementioned concerns!
>
> Your suggestion to evaluate using SVG with LLMs on more real-world datasets is greatly appreciated. We further studied the limitations of using SVG with LLMs in handling complex real-world tasks in the following.
>
>
> __1)__
>
> Due to limited computational resources, we performed experiments involving a subset of ImageNet. Specifically, we conduct experiments on [Imagenette](https://github.com/fastai/imagenette), which is composed of images from 10 classes in ImageNet. We achieved 68.14\% top-1 accuracy on the corresponding 10-class test set after training with SVG format using Vicuna-7B model, whereas a ResNet50 model receives 90.62\% accuracy on the test set after trained with 200 epochs.
>
>
> Our initial results suggest that LLMs can somehow understand the semantics of the real-world image by leveraging SVG, yet lag behind raster representations using the vision model architectures like ResNet with a clear gap. We believe it's mainly because of the loss of fine-grained details during SVG conversion. We have added your suggested experiments to our paper to further underscore the limitation of using SVG with LLMs in understanding complicated visual tasks, and we are committed to expanding our experiments with more categories and varied datasets in the final version.
>
>
> __2)__
>
> Additionally, we agree we’ve selected some datasets/tasks that might favor SVG, but we observed that even with the advanced large vision model ConvNext, performance on the simple CMNIST dataset (Table 2) deteriorates significantly under out-of-distribution conditions. Utilizing the SVG format, in contrast, might offer a complementary solution to large vision models for better handling of such out-of-distribution scenarios.

---

### Meta-Review · Program_Chairs · 2023-12-08

**Metareview:**

This paper investigates the use of Scalable Vector Graphics (SVG) in conjunction with Large Language Models (LLMs) for image understanding. It explores the conversion of images into XML-based textual descriptions via SVG, enabling LLMs to perform tasks that involve both discriminative and generative visual understanding. These tasks include visual reasoning, image classification under various conditions, image generation, and editing. The study demonstrates that LLMs exhibit a reasonable capability in these areas, particularly in comparison to some expert models.


The approach is simple and effective. The experiments are well conducted. However the reviewers left concerns, including lack of novelty, lack of in-depth ablation study, and insufficient comparison with multimodal LLMs.
An additional main concern shared across the reviewers is the generalization and applicability to real images for dealing with photographic content.
Although the authors put good efforts in the rebuttal, the reviewer still left the concern on the scope of this work including its experiment design not suitable for evaluating LLMs’ visual understanding ability.

**Justification For Why Not Higher Score:**

Using SVG with LLMs for visual tasks specifically targets limited scopes, rather than handling complicated images and photographic contents, which may limit the field of interest. The authors need to specify the scope of this work more clearly. The authors are encouraged to improve the paper and submit it to another conference.

**Justification For Why Not Lower Score:**

N/A

---

### Decision · Program_Chairs · 2024-01-16

Reject